No benefit of higher protein dosing in critically ill patients: a systematic review and meta-analysis of randomized controlled trials

Qin Yonggen 1
Huang Jian 2
Ping Xiaofeng 2
Zheng Hui 1
Zhang Kai 3
Xu Xiaoya 4
Yu Jiuqing 2 496662452@qq.com
1 Department of Emergency Medicine, Hangzhou Ninth People’s Hospital , Hangzhou , China
2 Department of Critical Care Medicine, Hangzhou Ninth People’s Hospital , Hangzhou , China
3 Department of Critical Care Medicine, Second Affiliated Hospital of Zhejiang University School of Medicine , Hangzhou, Zhejiang , China
4 Department of General Surgery, Lishui People’s Hospital , Lishui , China
Loenneke Jeremy
Electronic publication date: 2024 May 21
Publication date: 2024
Volume: 12
Electronic Location ID: e17433
Received 2024 Jan 25; Accepted 2024 Apr 29
Copyright: © 2024 Qin et al.
Copyright year: 2024
Copyright holder: Qin et al.
License: This is an open access article distributed under the terms of the Creative Commons Attribution License, which permits unrestricted use, distribution, reproduction and adaptation in any medium and for any purpose provided that it is properly attributed. For attribution, the original author(s), title, publication source (PeerJ) and either DOI or URL of the article must be cited.
License URL: https://creativecommons.org/licenses/by/4.0/

Keywords: Nutrition support, Protein supplementation, Critically ill patients, Intensive care unit, Meta-analysis

Funding: The authors received no funding for this work.

==============================
Purpose

The optimal range of protein dosage and effect of high-dose protein on critically ill patients remain controversial. We conducted a meta-analysis to compare higher and lower doses of protein supplementation for nutritional support in critically ill patients.

Methods

We searched the PubMed, Embase, Scopus, and Cochrane Library databases for randomized controlled trials that compared higher (≥1.2 g/kg per day) versus lower (<1.2 g/kg per day) doses of protein supplementation among critically ill adult patients. This search spanned from the inception of relevant databases to November 20, 2023. Our primary endpoint of interest was overall mortality, while secondary endpoints included length of stay in the intensive care unit, length of hospital stay, duration of mechanical ventilation, and incidence of acute kidney injury.

Results

Seventeen studies including 2,965 critically ill patients were included in our meta-analysis. The pooled analyses showed no significant difference in overall mortality (RR 1.03, 95%CI [0.92–1.15], P = 0.65, I2 = 0%), length of intensive care unit stay (MD 0.19, 95%CI [−0.67 to 1.04], P = 0.66, I2 = 25%), length of hospital stay (MD 0.73, 95%CI [−1.59 to 3.04], P = 0.54, I2 = 27%), duration of mechanical ventilation (MD −0.14, 95%CI [−0.83 to 0.54], P = 0.68, I2 = 8%), and incidence of acute kidney injury (RR 1.11, 95%CI [0.87–1.41], P = 0.38, I2 = 0%) between critically ill patients receiving higher or lower doses of protein supplementation.

Conclusions

For critically ill patients, the protein supplementation dose had no significant effect on clinical outcomes, including overall mortality, length of intensive care unit and hospital stay, duration of mechanical ventilation, and incidence of acute kidney injury.

Introduction

Critical illness is characterized by pronounced alterations in protein metabolism across multiple organs, primarily driven by the action of hormones and cytokines (Weijs et al., 2014; Wischmeyer, 2007). These perturbations manifest as heightened rates of protein breakdown concomitant with diminished rates of protein synthesis (Preiser et al., 2015; Wischmeyer, 2007). Survival rates among individuals suffering from critical illnesses have demonstrated notable improvements over the past few decades. Survivors frequently experience the onset of protein-energy malnutrition accompanied by a decline in muscle mass, which is concomitant with adverse clinical outcomes (Puthucheary et al., 2013). The critically ill patients often have multiple organ failure, metabolic disorders, endocrine dysfunction, and major hyper catabolism with severe muscle wasting (Hermans & Van den Berghe, 2015; Puthucheary et al., 2013). Furthermore, several therapeutic interventions, such as mechanical ventilation (MV) and long-term sedation during the intensive care unit (ICU) stay, can further exacerbate this process of muscle degradation (Friedrich, Diermeier & Larsson, 2018), leading to a substantial loss of muscle mass of up to 2% per day (Wandrag et al., 2019). The deleterious impact of malnutrition on patient prognosis is well documented, including increased mortality rates (Felder et al., 2015; Hiura, Lebwohl & Seres, 2020), higher rates of infection (Casaer et al., 2011), and prolonged durations of ICU and hospital stay (Casaer et al., 2011; Hiura, Lebwohl & Seres, 2020). Conversely, nutritional intervention has emerged as a potent modality for ameliorating clinical outcomes and mitigating morbidity and mortality among individuals at nutritional risk (Gomes et al., 2019; Schuetz et al., 2019). Furthermore, a subgroup analysis of a cluster randomized controlled trial (RCT) identified 242 critically ill patients at high risk of renal dysfunction at study entry and found that patients randomized to receive higher daily protein intake were significantly less likely to require renal replacement therapy (Doig et al., 2009).

Unfortunately, the optimal protein supplementation dose for critically ill patients in the ICU remains debatable. Existing RCTs investigating various protein dosage regimens have encountered limitations stemming from inadequate statistical power or inconsistent and substantial distinctions in protein dosing between groups (Patel et al., 2020). Limited by the weak evidence, nutrition societies worldwide recommend a diverse spectrum of protein dosages, spanning from 1.2 to 2.0 g/kg per day (Elke et al., 2019; McClave et al., 2016; Singer et al., 2019; Sioson et al., 2018). Notably, specific subpopulations of critically ill patients, including those with obesity, burn injuries, or trauma, are advised to consider even more elevated protein intakes, ranging from 2.0 to 2.5 g/kg per day (Elke et al., 2019; McClave et al., 2016; Singer et al., 2019; Sioson et al., 2018). Synthesizing high-quality research evidence to determine the appropriate dose of protein represents a top priority and substantial challenge for the critical care community (Arabi et al., 2017). Recently, Heyland et al. (2023) completed the EFFORT Protein trial with a large sample size to investigate the effect of different protein doses in critically ill patients. The EFFORT Protein trial (Heyland et al., 2023) compared high-dose (≥2.2 g/kg per day) with the usual dose (≤1.2 g/kg per day) of protein. These findings suggest that higher dosages of protein had no significant effect on mortality, but a lower dose of protein was associated with an expedited recovery and a diminished incidence of complications (Heyland et al., 2023). Consequently, we performed this meta-analysis to compare the effects of higher and lower protein doses on clinical outcomes in critically ill patients.

Methods

This meta-analysis was performed according to the updated Preferred Reporting Items for Systematic Reviews and Meta-Analyses (PRISMA) statement (Page et al., 2021) (see Supplemental Material S1). The research protocol was registered in advance in the Open Science Framework (https://osf.io/2yp86). Two independent authors (Yonggen Qin, Jian Huang) searched PubMed, Embase, Scopus, and Cochrane Library databases for relevant studies up to November 20, 2023. A literature search was conducted using keywords containing “proteins”, “amino acids”, “critically ill”, “ICU”, and “randomized”. The full search strategy is provided in Supplemental Material S2.

Eligibility criteria

Studies that fulfilled the inclusion criteria were as follows: (1) Type of study: RCTs. (2) Population: critically ill adult patients. (3) Intervention and comparison: the use of higher versus lower protein delivery through all routes of administration. The higher dose of protein supplementation was defined as actual delivered ≥1.2 g/kg per day, and lower dose of protein supplementation was actual delivered <1.2 g/kg per day, the threshold of protein dose was set according to the existing guidelines (Singer et al., 2019). (4) Outcomes: The primary outcome was overall mortality, including ICU mortality, in-hospital mortality, 28/30-day mortality, etc. If one study reported more than one kind of mortality, the mortality with the longest follow-up time was evaluated. Secondary outcomes included the length of intensive care unit stay, length of hospital stay, duration of MV, and incidence of acute kidney injury (AKI). AKI was defined according to the criteria of Kidney Disease Improving Global Outcomes (KDIGO) (Ostermann et al., 2020), or defined by the original authors.

Data extraction and quality assessment

Relevant studies were systematically retrieved and their characteristics, including attributes such as authorship, publication years, study design, sample size, demographic characteristics of the population, prescribed protein targets, administration modalities, intervention durations, and routes, as well as outcome measures, were extracted by two independent authors (Yonggen Qin, Jian Huang). In instances where studies reported continuous outcomes in the form of median and interquartile range, we used the median, interquartile range, and sample size to estimate the approximate mean value and standard deviation. The calculating formula were proposed by Wan et al. (2014), they also developed a software to estimate the mean value and standard deviation.

The methodological quality of the included studies was independently assessed by two authors (Xiaofeng Ping, Hui Zheng), utilizing the Cochrane risk of bias tool (Higgins et al., 2011). In instances of discordant assessments, consensus was achieved through a deliberative process involving inclusion of a third adjudicator (Jiuqing Yu).

Statistical synthesis and analysis

All statistical analyses and assessments of bias risk were conducted by Review Manager Version 5.3 and “meta” package within the R software environment (version 4.3.1). Considering the expected clinical heterogeneity in dose and duration of protein between the included trials, we used a random-effects model to compute the pooled risk ratio (RR) with corresponding 95% confidence intervals (CI) for dichotomous outcomes and mean difference (MD) with 95%CI for continuous outcomes. The assessment of heterogeneity among the individual studies was predicated on Higgins inconsistency (I2) statistics (Higgins et al., 2003), with substantial heterogeneity defined as an I2 value exceeding 30%. In addition, potential publication bias was evaluated by employing both funnel plot analysis and Egger’s regression test (Egger et al., 1997). If publication bias was identified, the trim-and-fill method (Duval & Tweedie, 2000) was applied to further assess the possible effect of publication bias in our meta-analysis. This method considers the possibility of hypothetical “missing” studies that might exist, imputes their results, and recalculates a pooled estimate that incorporates the hypothetical missing studies as though they actually existed.

To elucidate the potential origins of heterogeneity, we performed a predefined subgroup analysis stratified by the route of protein administration (enteral or parenteral route), as well as the study duration (≤14 days versus >14 days). Additionally, a sensitivity analysis was performed, systematically excluding each individual study to discern the influence of specific studies on the overall outcomes.

Results

Study identification and characteristics

The preliminary literature search yielded a cumulative total of 9,505 articles retrieved from multiple sources, including PubMed (n = 1,558), Embase (n = 2,353), Scopus (n = 3,318), and the Cochrane Library (n = 2,276). Within this pool, 3,779 entries were identified as duplicates and were subsequently excluded. Following the screening of abstracts, an additional 5,669 studies were deemed ineligible for inclusion. Subsequently, through an extensive evaluation of the full-text content, additional 40 studies were excluded for various reasons. Finally, 17 RCTs (Allingstrup et al., 2017; de Azevedo et al., 2019; Badjatia et al., 2020; Carteron et al., 2021; Chapple et al., 2021; Danielis et al., 2019; Doig et al., 2015; Dresen et al., 2021; Fetterplace et al., 2018; Heyland et al., 2023; Nakamura et al., 2021; Rugeles et al., 2013; Singer, 2007; van Zanten et al., 2018; Xiong & Qian, 2021; Yeh et al., 2020; Zhang et al., 2021) fulfilled the stipulated inclusion criteria were included in this meta-analysis. A comprehensive flowchart illustrating the detailed progression of the study selection process is presented in Fig. 1.

Figure 1 PRISMA 2020 flow diagram for this meta-analysis.

A comprehensive description of the features characterizing the studies incorporated in this analysis is presented in Table 1. A total of 2,965 patients were analyzed, with 1,482 patients receiving higher-dose protein regimens and 1,483 patients receiving lower-dose protein interventions over the respective study durations. The number of patients in each study ranged from 14 to 1,301. Fifteen studies had a relatively small sample size (<100 patients per arm), and the remaining two studies enrolled >400 patients. Different routes and doses of protein administration were also identified, and the protein was administered through the enteral route in 15 trials and two used parenteral nutrition (PN) or intravenous (IV) amino acids. Among the studies that used the enteral nutrition (EN) strategy, supplementary PN was allowed in seven studies. Sixteen studies reported weight-based protein delivery, the mean or median protein delivery ranged from 1.2 to 2.0 g/kg per day for the higher protein group, 0.5 to 1.1 g/kg per day for the lower protein group. Singer (2007) reported protein delivery in grams per day: the mean protein delivery was 150 g per day for the high-protein group and 75 g per day for the low-protein group. The difference of protein delivery between higher and lower protein group ranged from 0.2 to 1.0 g/kg per day.

Table 1 Characteristics of included studies.

Study and year	Design	N (higher/lower)	Population	Protein target	Protein delivered	Intervention period and route	Outcomes	
Heyland et al. (2023)	Single-blinded, multicenter	645/656	Adult patients (≥18 years) within ICU admission, expected to remain mechanically ventilated ≥48 h with nutritional risk factors	Higher: ≥2.2 g/kg per day;	Higher: 1.6 g/kg per day;	Started within 96 h of ICU admission and continued for up to 28 days, through EN/PN	60-day mortality, length of ICU stay, duration of MV, length of hospital stay	
Lower: ≤1.2 g/kg per day	Lower: 0.9 g/kg per day	
Zhang et al. (2021)	Single-blinded, single-center	21/20	Adult patients ≥18 years old with respiratory failure required MV for ≥7 days	Higher: 2.0 g/kg per day;	Higher: 1.7 g/kg per day;	Started within 24 to 48 h of ICU admission and continued for up to 5 weeks days, through EN	ICU mortality, length of ICU stay, duration of MV, AKI	
Lower: 1.2 g/kg per day	Lower: 1.0 g/kg per day	
Dresen et al. (2021)	Single-blinded, single-center	21/21	Adult patients (≥18 years) admitted to ICU and required MV	Higher: 1.8 g/kg per day;	Higher: 1.5 g/kg per day;	Started at ICU admission and continued for up to 28 days, through EN/PN	ICU mortality, length of ICU stay, duration of MV, AKI	
Lower: 1.2 g/kg per day	Lower: 1.0 g/kg per day	
Xiong & Qian (2021)	Open-label, single-center	26/27	Patients aged 18 to 70 years with severe traumatic brain injury in ICU	Higher: 1.2 to 1.7 g/kg per day;	Higher: ≥1.2 g/kg per day;	Started at ICU admission and continued for 7 days, through EN	28-day mortality	
Lower: 0.5 to 0.7 g/kg per day	Lower: ≤0.7 g/kg per day	
Carteron et al. (2021)	Open-label, single-center	100/95	Brain-injured patients (≥18 years) with GCS ≤8 and expected duration of MV ≥48 h	Higher: 9.4 g of proteins per 100 ml of formula;	Higher: 1.3 g/kg per day;	Started within 36 h of ICU admission and continued for up to 10 days, through EN	60-day mortality, length of ICU stay, duration of MV	
Lower: 7.5 g of proteins per 100 ml of formula	Lower: 1.1 g/kg per day	
Nakamura et al. (2021)	Single-blinded, single-center	60/57	Patients (≥20 years) admitted to ICU	Higher: 1.8 g/kg per day;	Higher: 1.5 g/kg per day;	Started within 48 h of ICU admission and continued for up to 10 days, through EN/PN	28-day mortality, length of ICU stay, duration of MV, length of hospital stay	
Lower: 0.9 g/kg per day	Lower: 0.8 g/kg per day	
Chapple et al. (2021)	Double-blinded, multicenter	58/58	Adult patients (≥18 years) admitted to ICU and required MV	Higher: 10 g of proteins per 100 ml of formula;	Higher: 1.5 g/kg per day;	Started within 48 h of ICU admission and continued for up to 28 days, through EN	90-day mortality, length of ICU stay, length of hospital stay, AKI	
Lower: 6.3 g of proteins per 100 ml of formula;	Lower: 1.0 g/kg per day	
Badjatia et al. (2020)	Open-label, single-center	12/13	Adult patients (≥18 years) with aneurysmal subarachnoid hemorrhage admitted to Neurocritical Care Unit	Higher: 1.8 g/kg per day;	Higher: 1.5 g/kg per day;	Started within 24 h of admission and continued for up to 12 days, through EN	Length of ICU stay	
Lower: 1.2 g/kg per day	Lower: 0.9 g/kg per day	
Danielis et al. (2019)	Open-label, single-center	19/21	Adult patients (≥18 years) admitted to ICU and required MV	Higher: protein fortified diet;	Higher: 1.8 g/kg per day;	Started at ICU admission and continued for a median of 7 days, through EN/PN	In-hospital mortality, duration of MV	
Lower: standard diet	Lower: NR	
de Azevedo et al. (2019)	Open-label, multicenter	57/63	Adult patients (≥18 years) admitted to ICU and required MV, with an expectation of stay ≥2 days	Higher: 2.0–2.2 g/kg per day;	Higher: 1.7 g/kg per day;	Started within 72 h of ICU admission and continued for up to 14 days, through EN/PN	In-hospital mortality, length of ICU stay, duration of MV	
Lower: 1.4–1.5 g/kg per day	Lower: 1.1 g/kg per day	
Yeh et al. (2020)	Open-label, single-center	19/17	Adult patients admitted to surgical ICU who were expected to require MV for >24 h	Higher: protein fortified diet;	Higher: 1.2 g/kg per day;	Started at ICU admission and continued for a median of 12 days, through EN	In-hospital mortality	
Lower: standard of care	Lower: 0.9 g/kg per day	
van Zanten et al. (2018)	Double-blinded, multicenter	22/22	Mechanically ventilated patients (aged ≥18 years) admitted to ICU, expected to require enteral nutrition for ≥5 days	Higher: 8 g of proteins per 100 kcal;	Higher: 1.5 g/kg per day;	Started within 48 h of ICU admission and continued for up to 10 days, through EN/PN	28-day mortality, length of ICU stay, duration of MV, length of hospital stay	
Lower: 5 g of proteins per 100 kcal	Lower: 0.8 g/kg per day	
Fetterplace et al. (2018)	Single-blinded, single-center	30/30	Adult patients ≥18 years in ICU, required MV ≥72 h	Higher: 1.5 g/kg per day;	Higher: 1.2 g/kg per day;	Started within 24 h of ICU admission and continued for up to 15 days, through EN	60-day mortality, length of ICU stay, duration of MV, length of hospital stay, AKI	
Lower: 1.0 g/kg per day	Lower: 0.8 g/kg per day	
Allingstrup et al. (2017)	Open-label, single-center	100/99	Patients ≥18 years required MV, with an expected ICU stay of ≥3 days	Higher: ≥1.5 g/kg per day;	Higher: 1.5 g/kg per day;	Started within 24 h of ICU admission and continued for up to 90 days, through EN/PN	180-day mortality, length of ICU stay, length of hospital stay, AKI	
Lower: standard care	Lower: 0.5 g/kg per day	
Doig et al. (2015)	Open-label, multicenter	239/235	Adult patients (≥18 years) admitted to ICU with an expectation of stay ≥2 days	Higher: supplement of 100 g of amino acids per day	Higher: 2.0 g/kg per day;	Started within 24 to 48 h of ICU admission and continued for up to 7 days, through PN	90-day mortality, length of ICU stay, duration of MV, length of hospital stay, AKI	
Lower: standard care	Lower: NR	
Rugeles et al. (2013)	Single-blinded, single-center	45/43	Adult patients (≥18 years) admitted to ICU with an expected period of EN ≥96 h	Higher: ≥1.5 g/kg per day;	Higher: 1.4 g/kg per day;	Started within 48 h of ICU admission and continued for up to 7 days, through EN	Length of ICU stay, duration of MV	
Lower: standard care	Lower: 0.8 g/kg per day	
Singer (2007)	Open-label, single-center	8/6	Critically ill patients required MV and parenteral nutrition	Higher: 150 g per day;	Higher: 150 g per day;	Started within 48 h of ICU admission and continued for up to 3 days, through PN	In-hospital mortality, AKI	
Lower: 75 g per day	Lower: 75 g per day	
Note:

ICU, Intensive Care Unit; EN, enteral nutrition; PN, parenteral nutrition; NR, not reported; g, gram; kg, kilogram; AKI, acute kidney injury; ml, milliliter; N, number of participants; kcal, kilocalorie.

Quality assessment

The appraisal of study quality, executed through the application of the Cochrane risk-of-bias tool (Fig. 2), showed that seven of the included studies failed to provide comprehensive details pertaining to random sequence generation and/or allocation concealment. Additionally, a pronounced susceptibility to performance bias was discerned, predominantly attributed to the absence of blinding measures encompassing participants and personnel in all but two studies. Concerning the blinding technique for outcome assessment, eight studies exhibited an indistinct risk of bias, potentially leading to either underestimation or overestimation of true treatment effects.

Figure 2 Assessment of quality by the Cochrane risk of bias tool.

Furthermore, we evaluated potential publication bias by employing both Egger’s test and funnel plot analysis (Supplemental Material S3: Fig. S1). The results detected a potential publication bias for overall mortality (Egger’s test, P < 0.05). Consequently, the trim-and-fill method was employed, and the results after imputing showed that there was no difference between the higher and lower protein dosing groups (RR 1.05, 95%CI [0.93–1.17], I2 = 0%, Supplemental Material S3: Fig. S2).

Outcomes

All concerned outcomes are summarized in Table 2.

Table 2 Outcomes of this meta-analysis.

Outcome	N	Results	
Overall mortality	15	RR 1.03, 95%CI [0.92–1.15], P = 0.65, I2 = 0%	
EN	13	RR 1.04, 95%CI [0.92–1.17], P = 0.51, I2 = 0%	
PN	2	RR 0.90, 95%CI [0.63–1.30], P = 0.58, I2 = 0%	
		Test for subgroup difference: P = 0.47, I2 = 0%	
≤14 days	9	RR 0.94, 95%CI [0.76–1.17], P = 0.60, I2 = 0%	
>14 days	6	RR 1.06, 95%CI [0.93–1.22], P = 0.38, I2 = 0%	
		Test for subgroup difference: P = 0.36, I2 = 0%	
Length of ICU stay	13	MD 0.19, 95%CI [−0.67 to 1.04], P = 0.66, I2 = 25%	
EN	12	MD −0.03, 95%CI [−1.04 to 0.97], P = 0.95, I2 = 20%	
PN	1	MD 0.90, 95%CI [−0.16 to 1.96], P = 0.10	
		Test for subgroup difference: P = 0.21, I2 = 36%	
≤14 days	7	MD 0.01, 95%CI [−1.06 to 1.08], P=0.99, I2=15%	
>14 days	6	MD 0.58, 95%CI [−1.15 to 2.32], P = 0.51, I2 = 43%	
		Test for subgroup difference: P = 0.58, I2 = 0%	
Length of hospital stay	7	MD 0.73, 95%CI [−1.59 to 3.04], P = 0.54, I2 = 27%	
EN	6	MD 0.31, 95%CI [−3.37 to 3.99], P = 0.87, I2 = 38%	
PN	1	MD 1.20, 95%CI [−1.48 to 3.88], P = 0.38	
		Test for subgroup difference: P = 0.70, I2 = 0%	
≤14 days	3	MD −0.76, 95%CI [−6.15 to 4.63], P = 0.78, I2 = 45%	
>14 days	4	MD 1.28, 95%CI [−2.56 to 5.11], P = 0.51, I2 = 34%	
		Test for subgroup difference: P = 0.55, I2 = 0%	
Duration of MV	10	MD −0.14, 95%CI [−0.83 to 0.54], P = 0.68, I2 = 8%	
≤14 days	6	MD −0.87, 95%CI [−1.78 to 0.05], P = 0.06, I2 = 0%	
>14 days	4	MD 0.51, 95%CI [−0.28 to 1.30], P = 0.21, I2 = 0%	
		Test for subgroup difference: P = 0.03, I2 = 80%	
Incidence of AKI	7	RR 1.11, 95%CI [0.87–1.41], P = 0.38, I2 = 0%	
EN	5	RR 1.16, 95%CI [0.81–1.67], P = 0.41, I2 = 0%	
PN	2	RR 1.07, 95%CI [0.78–1.48], P = 0.67, I2 = 0%	
		Test for subgroup difference: P = 0.74, I2 = 0%	
≤14 days	2	RR 1.07, 95%CI [0.78–1.48], P = 0.67, I2 = 0%	
>14 days	5	RR 1.16, 95%CI [0.81–1.67], P = 0.41, I2 = 0%	
		Test for subgroup difference: P = 0.74, I2 = 0%	
Note:

N, number of studies; ICU, intensive care unit; RR, risk ratio; MD, mean difference; CI, confidence interval; AKI, acute kidney injury.

Mortality

Fifteen studies (n = 2,833) reported overall mortality, and no difference was found between the higher and lower protein dosing groups (RR 1.03, 95%CI, [0.92–1.15]; P = 0.65; I2 = 0%; Fig. 3). Based on the study duration (≤14 days versus >14 days) and the route of protein delivery (enteral or parenteral route), no significant differences in overall mortality were observed between the subgroups (Table 2; Supplemental Material S3: Fig. S3). Besides, we have performed additional subgroup analysis for stratified by different follow-up time (e.g. ICU mortality, in-hospital mortality, 28/30, 60, 90, 180-day mortality). There was no significant difference between different follow-up time groups (Supplemental Material S3: Fig. S4).

Figure 3 Forest plot showing the difference between higher versus lower protein dosing for overall mortality (Allingstrup et al., 2017; de Azevedo et al., 2019; Carteron et al., 2021; Chapple et al., 2021; Danielis et al., 2019; Doig et al., 2015; Dresen et al., 2021; Fetterplace et al., 2018; Heyland et al., 2023; Nakamura et al., 2021; Singer, 2007; van Zanten et al., 2018; Xiong & Qian, 2021; Yeh et al., 2020; Zhang et al., 2021).

There was no change in the direction of results in the sensitivity analysis that omitted every single study at a time, indicating good robustness (Supplemental Material S3: Fig. S5).

Length of ICU stay

Thirteen studies (n = 2,805) reported on the length of ICU stay. No significant difference was found between the higher and lower protein dosing groups in the overall analysis (MD 0.19, 95%CI [−0.67 to 1.04], P = 0.66, I2 = 25%, Fig. 4A) or subgroup analysis (Table 2; Supplemental Material S3: Fig. S6). There was no change in the direction of results in the sensitivity analysis that omitted every single study at a time, indicating good robustness (Supplemental Material S3: Fig. S7).

Figure 4 Forest plot showing the difference between higher versus lower protein dosing for (A) length of ICU stay, (B) length of hospital stay (Allingstrup et al., 2017; de Azevedo et al., 2019; Badjatia et al., 2020; Carteron et al., 2021; Chapple et al., 2021; Doig et al., 2015; Dresen et al., 2021; Fetterplace et al., 2018; Heyland et al., 2023; Nakamura et al., 2021; Rugeles et al., 2013; van Zanten et al., 2018; Zhang et al., 2021).

Length of hospital stay

Seven studies (n = 2,299) reported on the length of hospital stay. No significant difference was found between the higher and lower protein dosing groups in the overall analysis (MD 0.73, 95%CI [−1.59 to 3.04], P = 0.54, I2 = 27%, Fig. 4B) or subgroup analysis (Table 2; Supplemental Material S3: Fig. S8). There was no change in the direction of results in the sensitivity analysis that omitted every single study at a time, indicating good robustness (Supplemental Material S3: Fig. S9).

Duration of MV

Ten studies (n = 2,040) reported on the duration of MV. No significant difference was found between the higher and lower protein dosing groups in the overall analysis (MD −0.14, 95%CI [−0.83 to 0.54], P = 0.68, I2 = 8%, Fig. 5A) or subgroup analysis (Table 2; Supplemental Material S3: Fig. S10). There was no change in the direction of results in the sensitivity analysis that omitted every single study at a time, indicating good robustness (Supplemental Material S3: Fig. S11).

Figure 5 Forest plot showing the difference between higher versus lower protein dosing for (A) duration of MV,(B) incidence of AKI (Allingstrup et al., 2017; de Azevedo et al., 2019; Carteron et al., 2021; Chapple et al., 2021; Danielis et al., 2019; Doig et al., 2015; Dresen et al., 2021; Fetterplace et al., 2018; Heyland et al., 2023; Nakamura et al., 2021; Rugeles et al., 2013; Singer, 2007; van Zanten et al., 2018; Zhang et al., 2021).

Incidence of AKI

Seven studies (n = 944) reported on the incidence of AKI. The incidence of AKI in patients receiving a higher dose of protein was higher than that in the lower-dose protein group (22.7% vs. 20.3%), although the difference was not statistically significant (RR 1.11, 95%CI [0.87–1.41], P = 0.38, I2 = 0%, Fig. 5B). In the subgroup analysis, there was no significant difference in the incidence of AKI between the high and low protein dosing groups (Table 2; Supplemental Material S3: Fig. S12). There was no change in the direction of results in the sensitivity analysis that omitted every single study at a time, indicating good robustness (Supplemental Material S3: Fig. S13).

Post hoc analysis

In consideration of the wide range of protein delivery among included studies, we performed a post hoc subgroup analysis stratified by the difference of protein intake between higher and lower protein group (≤0.5 versus >0.5 g/kg per day). No subgroup differences were detected between studies (Supplemental Material 3: Figs. S14 to S18).

Discussion

In this updated meta-analysis of RCTs that compared higher versus lower protein supplementation, we found that a higher dose of protein supplementation for critically ill patients was not associated with a significant effect on overall mortality, length of ICU and hospital stay, duration of MV, or incidence of AKI. Moreover, the different route and duration of protein administration had no significant effect on outcomes.

Regarding the optimal protein supplementation dosage for critically ill patients, previous systematic reviews and meta-analyses have yielded disparate conclusions and provided divergent recommendations. Hoffer & Bistrian (2012) suggested that administration of protein within the range of 2.0 to 2.5 g/kg per day may be considered a safe and potentially optimal strategy for most critically ill patients. However, the recommendations were formulated with the awareness that they draw upon a limited corpus of evidence characterized by both meager volume and suboptimal quality. Conversely, Davies et al. (2017) conducted an exhaustive analysis encompassing 14 RCTs enrolling a total of 3,238 patients, elucidating that there exists no substantial correlation between the quantity of administered protein and mortality. It is imperative to acknowledge that the mean protein provision across the study groups, notably 1.02 g/kg per day in the higher protein cohort and 0.67 g/kg per day in the lower protein group, conspicuously falls short of the recommended threshold (≥1.2 g/kg per day) (Singer et al., 2019). In order to ascertain whether guideline-endorsed protein doses could confer enhanced clinical outcomes, Fetterplace et al. (2020) performed a comparative investigation involving two divergent protein dosage regimens (<1.2 versus ≥1.2 g/kg per day) within the critically ill patient cohort. However, due to the relatively small number of included studies, limited to only six RCTs comprising 511 patients, definitive conclusions remain elusive. Subsequently, van Ruijven et al. (2023) included 14 RCTs and 15 non-randomized studies, they found high protein provision of more than 1.2 g/kg in critically ill patients seemed to improve nitrogen balance, changes in muscle mass and 60-day mortality. However, given the limited ability to draw definitive conclusion about the benefits of high protein from non-randomized studies, the meta-analysis combined non-randomized studies with RCTs might reduce confidence in their conclusion. More RCTs are needed to verify their results. In a most recent meta-analysis by Lee et al. (2024) spanning 23 RCTs encompassing a total of 3,303 patients, the authors sought to evaluate the impact of divergent protein delivery levels while maintaining similar energy provisions among groups, concerning clinical outcomes within the critically ill patient population. Notably, although higher protein delivery was associated with the mitigation of muscle mass depletion, discernible improvements in overall mortality, length of stay in the ICU and hospital, and mechanical ventilation duration were conspicuously absent. It is pertinent to acknowledge that, Lee et al. (2024) aimed to compare higher versus lower protein delivery without limitation of protein dose for the higher protein group (e.g. a higher protein dose in one study is the lower protein dose in another study). The protein delivery doses in higher protein group from several RCTs included in the meta-analysis by Lee et al. (2024) were lower than 1.2 g/kg per day, which was significantly lower than other included studies. Thus, these RCTs were excluded in our study.

Incorporating the most recent RCTs, the present meta-analysis is the most up-to-date investigation of its type. Collectively, our findings align with and substantiate the outcomes documented in previous meta-analyses, thereby strengthening the existing body of evidence. Our results align well with the findings from recent RCTs, which suggest that higher protein supplementation might be harmful for critically ill patients. Patients in the high-dose protein group had higher rates of AKI, although the difference was not statistically significant. The relationship between protein intake and AKI is complex and can depend on various factors including the context of the individual’s health status, the source of protein, and the amount consumed. Higher protein intake can potentially increase the risk of AKI, especially in individuals with pre-existing kidney disease or compromised kidney function. Conversely, lower protein intake may also be associated with an increased risk of AKI, particularly in certain clinical contexts such as critical illness or malnutrition. In these situations, inadequate protein intake can lead to muscle breakdown and decreased synthesis of proteins involved in maintaining renal function, potentially predisposing individuals to AKI. In our meta-analysis, higher rates of AKI observed in the high-dose protein group may suggest a potential adverse effect of high protein intake on kidney health.

The EFFORT Protein trial (Heyland et al., 2023) found that a higher protein dose was associated with increased 60-day mortality in patients with AKI at ICU admission. Furthermore, post-hoc subgroup analysis of the Nephro-Protective Trial (Doig et al., 2015) found an interaction between higher protein intake, baseline kidney injury, and worse clinical outcomes (Zhu et al., 2018). The observation that the higher-dosage protein cohort exhibited an elevation in ureagenesis suggests a potential metabolic burden characterized by heightened protein-amino acid catabolism in patients with AKI, particularly in those presenting concomitant impairments in muscle protein synthesis (Heyland et al., 2023). Consequently, a preceding investigation advocated for a reduction in protein intake to a range of 0.8 to 1.0 g/kg per day for patients at risk or those afflicted with acute or chronic renal insufficiency (Bounoure et al., 2016). This recommendation arises from concerns surrounding the potential deleterious effects of high-protein diets within the patients’ demographic (Hahn, Hodson & Fouque, 2018). Notably, a secondary analysis conducted within the framework of the EFFORT trial, focusing on patients with chronic kidney disease, underscored the significance of impaired renal function as a robust predictor of therapeutic responsiveness to nutritional interventions (Bargetzi et al., 2021). Furthermore, AKI can also act as a confounding variable in studies examining the relationship between protein intake and other health outcomes. For example, if individuals in the high-dose protein group are more likely to develop AKI, this could potentially confound the observed effects of protein intake on other outcomes including mortality, length of stay in ICU and hospital. Since in most of the included studies, neither baseline renal functions nor outcomes of renal function subgroups were reported, the current amount of data cannot support a comprehensive analysis of the effect of higher protein doses on patients with or without renal injury. Hence, the precise determination of an optimal protein target remains a subject of ongoing debate, necessitating further investigation aimed at elucidating the disease-specific quantity and quality of proteins most conducive to therapeutic outcomes.

It is imperative to acknowledge that energy delivery exhibited notable variability across the diverse trials under consideration. The allocation of energy resources among participants receiving higher and lower protein dosages was not uniformly standardized, with a noteworthy proportion of the enrolled studies deviating from prevailing guideline recommendations (McClave et al., 2016). Although it is essential to recognize that the quantity of energy administered has the potential to confound the implications of protein delivery (Compher et al., 2017; Hoffer et al., 2017), an illuminating insight emerges from the TARGET trial. Specifically, this trial demonstrated that in comparison with conventional care, the introduction of energy-dense delivery failed to exert any discernible influence on mortality rates or other patient-centric outcomes when protein provisions remained at comparable levels (Arunachala Murthy et al., 2022; Chapman et al., 2018). In addition, future studies are needed to confirm these findings. Currently, several ongoing RCTs are investigating the optimal range of protein dosage and effect of protein supplementation in critically ill patients. The TARGET protein (ACTRN12621001484831) is undertaken in eight ICUs to compare the 1.2 to 2.0 g/kg per day of protein with usual dietary protein (Summers et al., 2023). The PRECISE trial (NCT04633421) aimed to compare the daily protein of 1.3 g/kg with of 2.0 g/kg in ICU patients (van Gassel et al., 2023).

Strengths and limitations

Our study has several notable strengths, including a comprehensive and extensive approach to study selection, rigorous inclusion criteria, and application of high-quality statistical methodologies. Notably, our investigation distinguishes itself from prior research endeavors through the assimilation of the most current RCTs, including the EFFORT Protein trial (Heyland et al., 2023). This large-scale trial, which involved more than 1,000 patients from various ICUs, showed significant improvements in methodology, including strict allocation concealment and better separation between groups for protein supplementation. Our study provides the latest evidence on protein delivery in critically ill patients and highlights the importance of rigorous methodologies in clinical trials. Moreover, considering that the clinical heterogeneity at different doses and routes of protein delivery could have affected the results, we performed subgroup analyses and provided evidence of different doses and routes of protein delivery. These findings provide important practical recommendations for protein supplementation in critically ill patients at nutritional risk.

However, the current meta-analysis has certain limitations. The primary constraint of this meta-analysis pertains to the predominantly moderate-quality, small-scale, and single-center nature of the included studies. It is noteworthy that 15 of the encompassed trials would typically be categorized as small studies (<100 patients per arm), which could potentially introduce a bias known as small study effect bias (Zhang, Xu & Ni, 2013). The EFFORT trial, which enrolled more than 1,200 critically ill patients, accounts for the majority of weight in our analyses. The EFFORT trial might have larger effects on the overall effect size estimates, precision of the estimate, heterogeneity, generalizability and external validity. However, the result of sensitivity analysis by omitted the EFFORT trial was similar to the original result.

Second, there was substantial variability in the ranges of protein intake, dosing, timing, and route of protein administration across the included trials as well as variability in patient types and severity of illness. Furthermore, the studies incorporated in our analysis exhibited disparate criteria for intervention discontinuation or resumption, which might have contributed to the heterogeneity in our findings. The precision of our estimates for outcomes such as the length of ICU and hospital stays and the duration of MV was hampered by significant heterogeneity, engendering notable uncertainty in these results.

Additionally, several studies reported continuous variables in the form of median and interquartile range, necessitating conversion to mean and standard deviation, a process that potentially introduced bias into our findings. The realm of clinical heterogeneity encompassing patient factors, preexisting nutritional status, and intervention duration might also have influenced our results. The dose, route, initiation time, and duration of intervention were not sufficiently standardized. This clinical heterogeneity caused by the aforementioned differences in clinical trials can also lead to statistical heterogeneity, which may further affect the credibility of the conclusions.

Furthermore, several studies exhibited a moderate or indistinct risk of bias, particularly pertaining to performance bias, given the inherent challenges of blinding patients and dieticians in dietary studies.

Conclusion

In conclusion, there is increasing evidence for RCTs showing that a higher dose of protein supplementation was not associated with a significant effect on overall mortality, duration of MV, length of stay in the ICU and/or hospital, and incidence of AKI. Individualized clinical decision-making is critical in the administration of protein to critically ill patients with nutritional risk, considering the patient’s condition, nutritional status, and potential adverse effects. Future studies are warranted to provide a more comprehensive understanding of the optimal range of protein dosage and the effects of protein supplementation in critically ill patients.

Supplemental Information

Supplemental Information 1 PRISMA checklist.

Supplemental Information 2 Searching strategies.

Supplemental Information 3 Publication bias assessment by funnel plot and Egger’s test, subgroup and sensitivity analyses.

Supplemental Information 4 Systematic Review and/or Meta-Analysis Rationale.

The rationale for conducting the systematic review/meta-analysis. The contribution that it makes to knowledge in light of previously published related reports, including other meta-analyses and systematic reviews.

Supplemental Information 5 Xiaoya Xu justification.

Additional Information and Declarations

Competing Interests

Author Contributions

Data Availability

The authors declare that they have no competing interests.

Yonggen Qin conceived and designed the experiments, performed the experiments, authored or reviewed drafts of the article, and approved the final draft.

Jian Huang conceived and designed the experiments, performed the experiments, authored or reviewed drafts of the article, and approved the final draft.

Xiaofeng Ping performed the experiments, analyzed the data, prepared figures and/or tables, and approved the final draft.

Hui Zheng analyzed the data, prepared figures and/or tables, and approved the final draft.

Kai Zhang analyzed the data, prepared figures and/or tables, authored or reviewed drafts of the article, and approved the final draft.

Xiaoya Xu conceived and designed the experiments, analyzed the data, authored or reviewed drafts of the article, and approved the final draft.

Jiuqing Yu conceived and designed the experiments, authored or reviewed drafts of the article, and approved the final draft.

The following information was supplied regarding data availability:

This is a Systematic Review/Meta-Analysis.

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
