# Peer review of "No benefit of higher protein dosing in critically ill patients: a systematic review and meta-analysis of randomized controlled trials"

_PeerJ, doi:10.7717/peerj.17433_

## Round 0.1 · original submission · Major Revisions

Reviewers found merit in the paper but have offered suggestions for improvement (e.g. providing additional clarity in methods and interpreting the findings within the limitations of the available literature, etc).

Reviewer 1 ·

Basic reporting

no comment

Experimental design

Methods
Lines 116-117
“we applied the methodology advocated by Wan et al.[23] to transform the data format into mean values and standard deviations (SD).”

Please explicitly state what this methodology is. This is very important for transparency.
Results
Lines 160-162
“One study reported protein delivery in grams per day; the mean protein delivery was 150 g per day for the high-protein group and 75 g per day for the low-protein group.”

Please explain how this was used in the analysis. Please cite which paper this was.

Validity of the findings

no comments

Additional comments

Thank you for the opportunity to review this manuscript. The authors sought to conduct a meta-analysis investigating the effects of protein dose of clinical outcomes, such as mortality. The authors report that increased protein did not have any effect of their main outcome. Overall this paper is very well written and the authors should be commended for their hard work. I have several minor comments that need to be addressed before this is ready for publication.

Reviewer 2 ·

Basic reporting

- Not all clinical outcomes were studies; please adjust title
- Please remove the sentences on ethical approval and informed consent on line 94 and 95, it is not applicable to this study
- Please replace the word ‘exhaustive’ by ‘extensive’ on line 145.

Experimental design

- If studies reported multiple mortality outcomes, the outcome with longest follow-up time was evaluated. This may be debatable as it has the longest time span from intervention,. If effect on mortality is expected, is it expected to be on the longest follow-up time?
- Was protein dose or actual delivered protein studied? Dose and delivered are both used, please clarify.
- The method does not report information about imputing in the meta-analysis, but title of Fig S2 does state that data was imputed. Some studies are analyzed twice (‘filled’) with different effect size, why? Please provide information in the methods.
- Inclusion criteria is 1.2 vs <1.2 g/kg protein, in Singer 2007, no g/kg is reported
- The EFFORT trial accounts for the majority of weight in your analyses, please elaborate how this affects your results.
- One of the outcomes is AKI, and the authors state that patients in the high-dose protein group had higher rates of AKI as. Is AKI expected to occur from either lower or higher protein? Yes, effects of lower or higher protein may be different in patients with AKI. But AKI is now descibred as both outcome and confounder.

Validity of the findings

- Ranges of protein intake are wide among groups, how may this affect outcomes?
- Recently, another meta-analysis was published (van Ruijven 2023) where preservation of muscle mass in the higher protein group was found; how about other clinical outcomes, please elaborate. Lee also published an updated meta-analysis in jan 2024.
- On line 248-249 it is mentioned that this meta-analysis is all-encompassing, but only a limited number of outcomes were analysed
- Timing of protein is not described, how does this affect outcomes?

---

## Round 0.2 · accepted · Accept

The authors addressed the reviewers concern. The reviewer did provide one more suggestion but the authors can decide whether they choose to do that in the proof stage (i would encourage them to).

Reviewer 1 ·

Basic reporting

no comment

Experimental design

no comment

Validity of the findings

no comment

Additional comments

Overall The authors adequately addressed my comments. However, i have one minor suggestion for clarity.

"The calculating formula were proposed by Wan et al.(Wan et al. 2014), they also developed a software to estimate the mean value and standard deviation."

Please state that you used the excel sheet provided from this paper.